# Myoferlin Contributes to the Metastatic Phenotype of Pancreatic Cancer Cells by Enhancing Their Migratory Capacity through the Control of Oxidative Phosphorylation

**DOI:** 10.3390/cancers11060853

**Published:** 2019-06-19

**Authors:** Gilles Rademaker, Brunella Costanza, Sandy Anania, Ferman Agirman, Naïma Maloujahmoum, Emmanuel Di Valentin, Jean Jacques Goval, Akeila Bellahcène, Vincenzo Castronovo, Olivier Peulen

**Affiliations:** 1Metastasis Research Laboratory, Giga Cancer, University of Liège, 4000 Liège, Belgium; g.rademaker@uliege.be (G.R.); brunellacostanza@gmail.com (B.C.); sandy.anania@uliege.be (S.A.); f.agirman@uliege.be (F.A.); naima.maloujahmoum@uliege.be (N.M.); a.bellahcene@uliege.be (A.B.); vcastronovo@uliege.be (V.C.); 2Viral Vector Platform, Giga Research, University of Liège, 4000 Liège, Belgium; edivalentin@uliege.be; 3Imaging Platform, Giga Research, University of Liège, 4000 Liège, Belgium; jean-jacques.goval@uliege.be

**Keywords:** myoferlin, pancreas cancer, mitochondria, oxidative phosphorylation, metastasis

## Abstract

Pancreatic ductal adenocarcinoma (PDAC) is one of the deadliest malignancies with an overall survival of 5% and is the second cause of death by cancer, mainly linked to its high metastatic aggressiveness. Accordingly, understanding the mechanisms sustaining the PDAC metastatic phenotype remains a priority. In this study, we generated and used a murine in vivo model to select clones from the human Panc-1 PDAC cell line that exhibit a high propensity to seed and metastasize into the liver. We showed that myoferlin, a protein previously reported to be overexpressed in PDAC, is significantly involved in the migratory abilities of the selected cells. We first report that highly metastatic Panc-1 clones expressed a significantly higher myoferlin level than the corresponding low metastatic ones. Using scratch wound and Boyden’s chamber assays, we show that cells expressing a high myoferlin level have higher migratory potential than cells characterized by a low myoferlin abundance. Moreover, we demonstrate that myoferlin silencing leads to a migration decrease associated with a reduction of mitochondrial respiration. Since mitochondrial oxidative phosphorylation has been shown to be implicated in the tumor progression and dissemination, our data identify myoferlin as a valid potential therapeutic target in PDAC.

## 1. Introduction

With a five-year survival lower than 5% and a steady increase of its incidence, pancreatic ductal adenocarcinoma (PDAC) is a major public health problem [1]. It is predicted that PDAC will become the second cause of death by cancer by 2030 [2]. To date, neither a performant early diagnostic marker nor a satisfactory treatment has been found for this disease [3]. Metastatic dissemination is the most frequent cause of death for PDAC patients, most of them being diagnosed after the primary tumor dissemination, leading to an extremely poor survival rate [4,5]. In consequence, the effort to better understand the precise molecular mechanism controlling the PDAC metastatic phenotype remains a major priority for cancer research.

Metabolism reprogramming was demonstrated to be compelling for tumor progression. In this process, cancer cells rely on glucose consumption via an increased uptake and glycolysis even in the presence of sufficient oxygen [6]. This process, named the Warburg effect, has been recognized for decades as the principal metabolism model for cancer cells. Metabolism reprogramming also affects the tumor microenvironment with the establishment of metabolic and oxidative cross talk between cancer and stromal cells represented by the reverse Warburg effect [7,8]. In the context of PDAC, cancer cell metabolism is also supported by amino acids produced by autophagy in stellate cells [9]. Thanks to the Warburg effect, other energy pathways (i.e., Krebs cycle and oxidative phosphorylation) are frequently shut down or reduced and mitochondria are thought to be restricted to the role of a biosynthetic hub [10]. According to the idea that metabolism switching to glycolysis is a hallmark of cancer [11], new therapeutic strategies have been developed and attempt to alter this specific pathway. However, recent studies have demonstrated that certain types of cancer cells, including PDAC cells, rely on oxidative phosphorylation (OXPHOS) to proliferate, survive, or resist treatment [12,13]. In consequence, glycolysis inhibitors are only efficient on few tumors or on few cells whereas the others switch to OXPHOS [14,15]. Concerning metastatic progression, several studies suggest that the Warburg effect is necessary to start the metastasis process [16] while others indicate that OXPHOS is required to allow the cancer cells to migrate [17,18]. To the best of our knowledge, nothing has ever been shown in the specific context of PDAC.

Myoferlin is a 230 kDa protein involved in different membrane fusion processes, including myoblast fusion into myotubes [19], endocytosis in endothelial cells [20], recycling of membrane receptors [21], exosome fusion to recipient cell [22], and exocytosis of growth factors [23]. Our laboratory has previously reported an overexpression of myoferlin in several cancer types including PDAC [24] where its expression is associated with a poor survival [25]. We previously showed the importance of myoferlin in the preservation of a metabolic flexibility [26] and more specifically in the control of mitochondrial function and structure [25,27]. Myoferlin, by a not yet fully understood mechanism, allows an optimized OXPHOS activity in pancreatic and colon cancer cells.

Interestingly, an elegant mathematical model predicts the role of myoferlin in the metastatic spreading of breast cancer [28]. This model has been partially validated in vitro, including by our laboratory, suggesting the inability of breast cancer cells to perform an epithelial-to-mesenchymal transition (EMT) when myoferlin is depleted [21,29]. Moreover, a pharmacological compound (WJ460, Chemical Abstracts Service registry number #1415251-36-3) was designed to interact with a specific C2 domain of myoferlin [30]. This molecule revealed a very promising anti-tumor activity in breast cancer and sharply reduced the incidence of lung metastasis.

Here, we report, for the first time, myoferlin as overexpressed in highly metastatic pancreatic cells generated by in vivo selection. We demonstrated that myoferlin overexpression leads to the migratory phenotype of the cells and is associated with an increased mitochondrial activity.

## 2. Results

### 2.1. Myoferlin Abundance Correlates with Pancreatic Cell Migratory Abilities and Oxygen Consumption Rate

We first evaluated the migration abilities of a collection of human pancreatic cell lines (Panc-1, BxPC-3, PaTu8988T, and MiaPaCa-2) in correlation with their respective myoferlin expression levels. Myoferlin expression levels were high in BxPC-3 and Panc-1 cell lines while they were low in PaTu8988T and MiaPaCa-2 cell lines (Figure 1A). The migration kinetics were assessed by a scratch assay (Appendix A) and showed a significant difference (*P* < 0.0001) in cell line migration speed (Figure 1B). Interestingly, we observed a high correlation (r = 0.9545, *P* = 0.023) between the wound-healing speed and the myoferlin abundance. Indeed, BxPC-3 were the highest myoferlin expressing cells and the fastest (28.3 ± 2.1 µm/h) to migrate while MiaPaCa-2 had the lowest myoferlin level and were the slowest (1.5 ± 0.3 µm/h) to migrate. At the end point (16 h after scratch), a significant difference remained between all cell lines (Figure 1C). We then performed Boyden’s chamber assay to evaluate 3D migration in the selected cell lines. Our results confirmed a significantly higher migration rate for BxPC-3 and Panc-1 cell lines in comparison with PaTu8988T and MiaPaCa-2 (Figure 1D and Appendix A). The highly migratory cell lines (BxPC-3 and Panc-1) presented the highest oxygen consumption rate (OCR) (Figure 1E), and a highly significant correlation was observed between myoferlin abundance and basal OCR (r = 0.9997, *P* = 0.0003).

### 2.2. Migration Is Dependent on OXPHOS in High Myoferlin Expressing PDAC Cell Lines

Encouraged by these unforeseen correlations and owing to our previous results showing the importance of myoferlin in the control of the mitochondrial function in PDAC, we decided to investigate the importance of OXPHOS in the migratory phenotype of the high myoferlin expressing cells BxPC-3 and Panc-1. OXPHOS was first impaired by a high concentration of mitochondrial respiratory chain uncoupler (carbonyl cyanide-*4*-(trifluoromethoxy)phenylhydrazone—FCCP) or by mitochondrial respiratory chain complex 3 inhibitor (antimycin A) mixed with ATP-synthase inhibitor (oligomycin). These compounds were previously reported to impair mitochondrial function in neuroblastoma SH-SY5Y cell line [31]. In BxPC-3 and Panc-1 cells the antimycin/oligomycin mix sharply decreased the OCR, indicating a strong impairment of the respiratory complexes. FCCP produced a slow reduction of the OCR, as described by Figarola and coworkers [32] with a higher FCCP concentration, indicating that the uncoupling activity of the compound was not able anymore to drive mitochondria to maximal OCR. Since this decrease was slower in Panc-1 cells than in BxPC-3, we decided to analyze the mitochondrial potential of Panc-1 cells after antimycin A + oligomycin or FCCP treatments. Both conditions reduced the tetramethylrhodamine, ethyl ester (TMRE) fluorescence, indicating a loss of the mitochondrial membrane potential and confirming the impairment of the mitochondrial function (Figure 2A). Both OXPHOS inhibitors significantly decreased (two to three-fold) the cell migration in a scratch assay (Figure 2B,C).

### 2.3. Myoferlin Is Required for PDAC Cell Migration and OXPHOS

We next inhibited myoferlin synthesis by using small interfering RNA (siRNA) and monitored 2D and 3D cell migration. Myoferlin silencing significantly reduced 2D cell migration in BxPC-3 (two-fold) and Panc-1 (three-fold) cell lines (Figure 3A). In the case of a myoferlin-silenced BxPC-3 cell line, we first observed a retraction of the wound margins explaining the specific shape of the migration kinetic curves. A similar amplitude of reduction was observed in the 3D Boyden’s chamber assay after myoferlin depletion (Figure 3B,C). Surprisingly, myoferlin silencing did not alter the abundance of the E-cadherin and vimentin EMT markers, suggesting that the migratory phenotype modification might be mainly a metabolic consequence (Appendix A). We then confirmed the impact of myoferlin silencing on OXPHOS. We showed in both cell lines a significant decrease of basal and maximal OCR when myoferlin was silenced (Figure 3D) while complex 1 (NADH:ubiquinone oxidoreductase subunit 5—NDUFB5) and 4 (cytochrome c oxidase subunit 4—COX IV) abundance was not altered (Appendix A).

### 2.4. Myoferlin Is Overexpressed in Cells with High Metastatic Potential

In order to enhance the relevance of our in vitro data, we undertook the development of an experimental liver metastases model of PDAC in immunodeficient mice. The model is based on three successive rounds of in vivo selection of liver-tropic Panc-1 cells (Figure 4A) inspired from colon cancer studies [33,34]. The in vivo selection method is a useful approach for the identification of biological mechanisms enhanced during liver metastasis [34]. We were able to produce three successive clones of Panc-1 cells. As expected, the in vivo selection progressively increased the metastatic potential of the cells collected in the liver (highly metastatic, HM) (Figure 4B). Indeed, according to round number, HM clones produced faster and more abundant liver metastatic foci. At the opposite, clones collected from the spleen (low metastatic, LM) progressively lost their metastatic potential (data not shown). The specific metastatic behavior of these clones was not the result of a different proliferative capability as suggested by a proliferation assay (Appendix A). Indeed, we showed a slight but significant increase of the proliferation rate in LM clones according to the number of the selection round while a decrease was observed in HM clones. We then evaluated vimentin, snail, and E-cadherin abundance in LM and HM clones (Figure 4C, quantification in Appendix A). It appeared that E-cadherin, an epithelial marker, was only detected in LM clones with a progressive increase of its expression according to the injection round while snail was not significantly altered at protein level. The mesenchymal marker vimentin was detected in all samples with a progressive decrease trend in LM clones and a progressive increase in HM clones (*P* = 0.030). These results were confirmed at mRNA level (Figure 4D). E-cadherin expression in EMT is under the repressive control of the snail transcription factor [35]. The mRNA expression of snail decreased progressively in LM clones while it increased in HM clones, confirming the EMT in HM clones. Satisfied with the characteristics of the HM and LM clones, we evaluated their myoferlin expression level. We demonstrated that myoferlin protein abundance (Figure 4C, quantification in Appendix A) and mRNA expression (Figure 4D) decreased in LM clones according to the round number while they increased in HM clones. Interestingly, the 180 kDa isoform of myoferlin, undetectable in LM clones, appeared in the HM clones. To further validate the link between myoferlin and metastatic potential, we took advantage of colon HT29 HM and LM clones generated by Price et al. [33]. Consistently, we found that myoferlin abundance was five-fold higher in HT29 HM than in LM (Appendix A) suggesting a link between myoferlin and in vivo metastasis potential.

### 2.5. Myoferlin Silencing Decreases OXPHOS Activity and Migration Ability in HM Clones

As myoferlin has been shown to be a key element to sustain a high metabolic flexibility [26] and an optimal mitochondrial fitness [25], we decided to compare the oxygen consumption rate (OCR) of HM and LM clones (Figure 5A). No significant differences were observed between LM and HM clones from the first selection round. Similarly, LM clones from the first selection round were not significantly different from the LM clones obtained after three rounds of in vivo selection. Interestingly, HM clones selected by three rounds of injection (HM3) showed a significantly higher OCR than HM clones selected by only one round (HM1) of injection, indicating a progressive amplification of OXPHOS according to the increase of the metastatic potential, resulting in an enhanced OCR in HM clones compared to that in LM clones in the last round of selection. A similar observation was made in HT29 LM and HM clones (Appendix A). To depict a more complete view of the clone metabolism, we estimated the gene expression of different metabolism-related proteins; we showed that the LM3 clone was globally more glycolytic with a significantly increased expression of glucose transporter (GLUT1), pyruvate kinase muscle 2 (PKM2), and lactate dehydrogenase (LDHA) whereas the HM3 clone was more oxidative with a higher expression of isocitrate dehydrogenase (IDH) (Figure 5B). Taken together, our results allow us to suggest that high OXPHOS is selected in HM clones to support their high metastatic potential. Surprisingly, difference was observed neither in the gene expression of glutamine-related enzyme (Figure 5B) nor in the cytochrome c oxidase abundance (Figure 5C, quantification in Appendix A). At the opposite, the abundance of the main positive regulator of OXPHOS PGC1-α followed the migratory potential of the LM and HM clones (Figure 5C, quantification in Appendix A). We next wanted to functionally link the increased OXPHOS activity of the HM clones with their high myoferlin expression. We thus silenced myoferlin in the HM3 clone and monitored the OCR. We observed a significant decrease of basal and maximal OCR following myoferlin silencing in the HM3 clone (Figure 5D) while no significant modification was observed in the LM3 clone (Appendix A). The same observation was made in HT29 LM and HM clones (Appendix A). We then wanted to provide a direct evidence of the requirement of myoferlin for migration. We thus silenced myoferlin in the HM3 clone and assessed its 2D migration ability. The HM3 clone transfected with irrelevant siRNA migrated slightly faster that the parental Panc-1 cells (respectively 13.2 ± 1.1 vs. 11.5 ± 0.3 µm/h) while the HM3 clone transfected with myoferlin siRNA#1 showed a significantly slower migration speed than the irrelevant siRNA-transfected HM3 clone (respectively 8.8 ± 0.7 vs. 13.2 ± 1.1 µm/h, *P* = 0.002) (Figure 5E), making them similar to PaTU8988T cell line (6.2 ± 0.1 µm/h).

## 3. Discussion

For decades, glycolysis has been shown to be the spearhead of metabolism in cancer cells. This principle, called the Warburg effect, is now accepted to be too simplistic to explain all metabolism features of cancer cells, and oxidative phosphorylation is now recognized as an important part of cancer cell fuel and energy provision. Moreover, it is also shown that mitochondria are involved in therapy resistance and metastases in different in vitro and in vivo models. Based on our previous publications demonstrating the relevance of myoferlin to maintain an optimized mitochondrial structure and function in pancreas cancer [25], we hypothesized that this protein is a key element for pancreatic cancer cell migration, a mandatory phenotype to successfully form metastases.

In the current study, we highlight for the first time a positive and causal correlation between myoferlin abundance in PDAC cell lines and their ability to migrate in 2D and 3D models. MiaPaCa-2 cells have a barely detectable myoferlin level and were the slowest to migrate while others reported significant migratory ability [36]. A major difference in the experimental setup can explain this apparent discrepancy. Indeed, Roy and coworkers coated the upper well of the Boyden’s chamber to evaluate MiaPaCa-2 migration while our experiments were performed without any coating. Controversy lies in the migratory ability of PDAC cell lines [37] but a recent report indicates a higher EMT potential in Panc-1 than in MiaPaCa-2 [38]. This phenotype specificity can arise from the relative expression of gene regulators such as HOX transcript antisense RNA (HOTAIR) long non-coding RNA (lncRNA) overexpressed in Panc-1 in comparison to MiaPaCa-2 and associated with enhanced cell invasion [39] and EMT [40]. Intriguingly, in a muscle repair context, muscle injuries induce an extensive regulation in coding and non-coding transcripts, among which myoferlin overexpression precedes the one of HOTAIR [41], suggesting a link between these factors or between myoferlin and other non-coding RNA.

Myoferlin silencing impaired the 2D and 3D in vitro migration ability of PDAC cell lines as previously suggested in breast cancer [26,30,42]. The present results are in accordance with the reduction of cellular ATP content observed after myoferlin silencing in Panc-1 [25] and show the importance of energy flexibility in PDAC progression [43]. In order to study the role of myoferlin in another context, we have elaborated an experimental metastasis model by in vivo selection of liver-tropic PDAC cells. This model relies on the observation of tumor or cell line heterogeneity, where highly metastatic cells are present as a subpopulation in the primary tumor [44] or cell line, including Panc-1 [38]. The progressive selection of these highly metastatic cells is probably the main driver for the establishment of the described model. The elaborated model solves a problem inherent to the difficulty of collecting pancreas–liver metastasis for research purposes since advanced PDAC patients are generally not eligible for surgery.

For the first time, we showed that myoferlin abundance parallels the in vivo metastatic potential of PDAC cells. We do believe that the in vivo selection is more relevant than an in vitro selection previously used to show that the mitochondrial switch promotes metastases [17]. Indeed, cancer cells would most probably use any available energy substrates to proliferate and spread. Consequently, in vivo selection seems more appropriate as it mimics the low glucose availability encountered in patients. Using this relevant PDAC metastases model, we were able to associate high myoferlin expression to OXPHOS and to a migratory phenotype. The main results were recapitulated in a similar colon model [33]. We can speculate on the mechanism by which myoferlin modulates OXPHOS. Of course, we cannot ignore our previous observation demonstrating the disorganization of the mitochondrial network in the absence of myoferlin [25]. Obviously, the connection between the bioenergetics and the mitochondrial morphology was previously established [45], but this only postpones the problem of determining the link between myoferlin and mitochondrial morphology. As the physiological function of myoferlin is related to membrane biology, it is likely that the mechanism by which myoferlin controls OXPHOS is also related to membrane processes among which we can propose iron uptake or mitochondrial fusion.

Our discovery increases the value of myoferlin as a potential therapeutic target. In this context, a synthetic pharmacological compound targeting myoferlin was recently described [30]. It appeared to have an efficient anti-metastatic activity at a nanomolar range in breast cancer. The metabolic feature of metastatic cells, meaning their mitochondrial switch from glycolysis to OXPHOS, could explain the proportion of early/small metastasis that cannot be detected by 18F-deoxyglucose-positron emission tomography (PET) [46]. As suggested by Halbrook and Lyssiotis, an 18F-labeled glutamine analog could potentially be used to identify and localize with more efficiency the early/small metastases [47].

## 4. Materials and Methods

### 4.1. Cells and Chemicals

Human pancreatic cancer cell lines BxPC-3 (ATCC CRL-1687), Panc-1 (ATCC CRL-1469), and MiaPaCa-2 (ATCC CRL-1420) were respectively a kind gift from Professor Bikfalvi (Inserm U1029, Bordeaux, France), Professors Burtea and Muller (NMR Laboratory, University of Mons, Belgium), and Professor De Wever (Laboratory of Experimental Cancer Research, University of Gent, Belgium). Authentication of these cell lines was done by short tandem repeat (STR) profiling (DSMZ, Braunschweig, Germany). Colon HT29 (low and high metastatic clones) was a generous gift from Professor Giavazzi (Mario Negri Institute for Pharmacological Research, Bergamo, Italy). PaTu8988T (DSMZ ACC162) was purchased from DSMZ (Braunschweig, Germany). All reagents were obtained from Sigma (Bornem, Belgium), except when mentioned otherwise. Antibodies were purchased from Santa Cruz Biotechnology (Santa Cruz, CA, USA): PGC1a (sc-13067), HSC70 (sc-7298); Sigma Life Sciences (Bornem, Belgium): Myoferlin (HPA014245), vimentin (V6389); Becton Dickinson Transduction Laboratories: E-cadherin (610181); Invitrogen (Carlsbad, CA): Cytochrome c oxidase (COX IV) (459600); or R&D Systems (Minneapolis, MN, USA): Snail (AF3639).

### 4.2. Cell Culture

BxPC-3 was maintained in RPMI1640 medium supplemented with 2.5 g/L glucose, 1 mM sodium pyruvate, and 10% fetal bovine serum (FBS). Panc-1 was cultured in Dulbecco’s modified Eagle’s medium (DMEM) supplemented with 10% FBS. MiaPaCa-2 was maintained in DMEM supplemented with 10% FBS, 1 mM sodium pyruvate, and 4 mM L-glutamine. PaTu8988T was cultured in DMEM supplemented with 5% FBS, 5% horse serum, and 2 mM L-glutamine. HT29 clones were maintained in RPMI1640 supplemented with 10% FBS. Cells were cultured in a 37 °C, 5% CO_2_ incubator. Cells were used between passages 1 and 10 and checked every month for mycoplasma.

### 4.3. Small Interfering RNA Transfection

Myoferlin silencing was performed by the transfection of 20 nM small interfering RNA (siRNA) targeting myoferlin (Eurogentec, Seraing, Belgium—siRNA#1 CCCUGUCUGGAAUGAGAUUTT; siRNA#2 CUGAAAGAGCUGUGCAUUATT). Irrelevant siRNA was targeting luciferase (Eurogentec, Seraing, Belgium—CUUACGCUGAGUACUUCGATT). HT29, Panc-1 cell lines and clones were transfected using calcium phosphate as described previously [48]. BxPC-3 were transfected with Lipofectamine (Life Technologies, Carlsbad, NM, USA) according to the manufacturer’s recommendations. All experiments were performed 48 h after transfection.

### 4.4. Scratch Assay

Scratch assay was performed using IncuCyte S3 (Sartorius, Ann Arbor, MI, USA). Cells were seeded at 20,000 cells/well the day before the experiment to achieve confluence the day of the experiment. Scratches were then applied to cells using a specific wound-maker tool. When needed, exogenous treatments (10 µM carbonyl cyanide-*4*-(trifluoromethoxy)phenylhydrazone—FCCP or 1 µM antimycin A/10 µM oligomycin mix) were applied immediately after scratching. To avoid proliferation bias, migration was only evaluated during 16 h.

### 4.5. Boyden’s Chamber Migration

Cells were detached using trypsin/EDTA and then counted using a CASY counter (OLS, Bremen, Germany). Cells were resuspended in DMEM with 1% bovine serum albumin (BSA) and 1% penicillin streptomycin. Cells were then seeded in Boyden’s chambers (200,000 cells/well). Chemoattractant (10% FBS) was placed in the bottom compartment. Cells were allowed to migrate for 24 h before fixation and coloration.

### 4.6. Western Blotting

Cell lysis was performed using sodium dodecyl sulfate (1% SDS) with protease and phosphatase inhibitors. Samples were then resolved by polyacrylamide gel electrophoresis as described previously [49].

### 4.7. Extracellular Flux Analysis

All experiments were performed with a Seahorse XFp extracellular flux analyzer (Agilent, Santa Clara, CA, USA) as described previously [25]. Cells were seeded 24 h before the assay at 20,000 cells/well density.

### 4.8. Mitochondrial Membrane Potential Assay

Panc-1 cells were resuspended in PBS then incubated during 15 min with 10 nM tetramethylrhodamine, ethyl ester (TMRE). PBS-washed cells were resuspended in PBS/0.2% BSA and analyzed by flow cytometry (FACS Fortessa, Becton Dickinson, Franklin Lakes, NJ, USA) at 526/575 nm wavelength.

### 4.9. In Vivo Selection of Liver-Tropic Cells

Panc-1 (10^6^ cells) were resuspended in DMEM then injected in the spleen of non-obese diabetic-severe combined immunodeficiency (NOD-SCID) mice. Four weeks after injection, the mice were sacrificed. Spleen and liver were both dissociated separately with a mix of collagenase and hyaluronidase (1600 mg/mL and 300 mg/L, respectively) in Hank’s Balanced Salt Solution (HBSS) for 30 min at 37 °C under agitation. Homogenates were then filtrated (70 µm mesh) and cells were seeded separately in a T25 flask with DMEM supplemented with 10% FBS, 1% penicillin streptomycin, and 0.4% fungizone. To eliminate fibroblasts, cells were kept during four passages and purity was checked under microscope. Resulting cells were re-injected separately according to the same procedure (Figure 4A). Three rounds of selection were performed allowing the selection of three successive clones of highly metastatic (HM) and of low metastatic (LM) cells, respectively, isolated from the liver and the spleen [33].

### 4.10. RNA Isolation and Reverse Transcription-PCR (RT-PCR)

RNA extraction, reverse transcription and PCR were performed as previously described [50]. A mix of cDNA, primers, probe (Universal ProbeLibrary System, Roche, Palo Alto, CA, USA) and 2× FastStart Universal Probe Master Mix (Roche, Palo Alto, CA, USA) was prepared and then PCR was performed using a 7300 Real Time PCR System and the corresponding manufacturer’s software (Applied Biosystems, Foster City, CA, USA). Relative gene expression was normalized to 18S rRNA. Three technical replicates of each sample were analyzed, and data are presented as mean ± SEM of three biological replicates.

### 4.11. Ethics

All animal experimental procedures were performed according to the Federation of European Laboratory Animal Sciences Associations (FELASA) and were reviewed and approved by the Institutional Animal Care and Ethics Committee of the University of Liège, Belgium (#18-2035).

### 4.12. Statistics

Results are reported as means with standard deviation (SD). Two-sided statistical analysis was performed using one-way or two-way ANOVA depending on the number of grouping factors. Unless mentioned otherwise, group means were compared by unpaired Student’s t-test or Bonferroni’s post-test according to the group number. Welch’s correction was applied when homoscedasticity was suspected. *P* < 0.05 was considered as statistically significant. All experiments were performed as several independent biological replicates. Statistics were performed using Prism v5.0f.

## 5. Conclusions

Myoferlin is an emerging pro-oncogenic protein. It appears from our previously publications [25,26,27] as well as from this original report that it is a key element of mitochondrial dynamic and/or function. Recent studies have demonstrated that several cancer cells, including PDAC cells, rely on OXPHOS to proliferate, survive, or resist treatment [12,13]. In this context, targeting genetically or pharmacologically myoferlin appeared as a sound and promising strategy in PDAC.

## Figures and Tables

**Figure 1 cancers-11-00853-f001:**
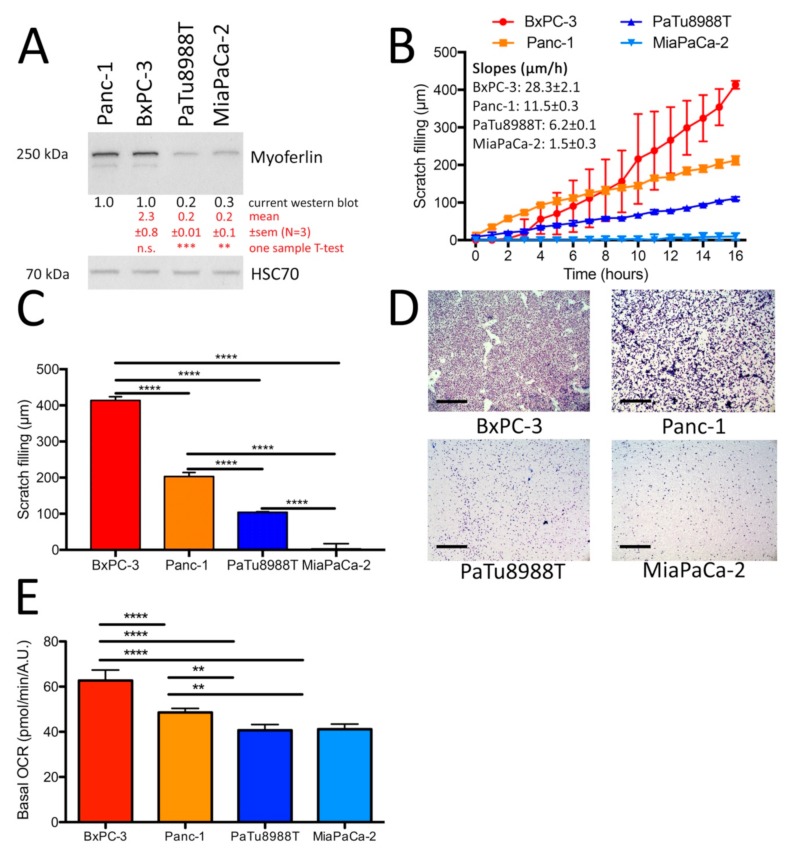
Myoferlin abundance is correlated with pancreatic cell migratory phenotypes and oxygen consumption rate. (**A**) Myoferlin relative abundance in Panc-1, BxPC-3, PaTu8988T, and MiaPaCa-2 cell lines. Heat Shock Cognate 71 kDa protein (HSC70) was used as an internal loading control. Myoferlin abundance was compared by one sample T-test to Panc-1 sample mean arbitrary fixed to 1. (**B**) Two-dimension migration kinetic assay (scratch assay) of BxPC-3, Panc-1, PaTu8988T, and MiaPaCa-2 cell lines. (**C**) End point (16 h) of two-dimension migration kinetic assay (scratch assay) of BxPC-3, Panc-1, PaTu8988T, and MiaPaCa-2 cell lines. (**D**) Three-dimension migration assay in Boyden’s chamber of BxPC-3, Panc-1, PaTu8988T, and MiaPaCa-2 cell lines (scale bar = 500 µm). (**E**) Basal oxygen consumption rate (OCR) in BxPC-3, Panc-1, PaTu8988T, and MiaPaCa-2 cell lines. One representative experiment out of three is illustrated. Each data point represents mean ± SD (± SEM for panel A), n = 3. **** *P* < 0.0001, *** *P* < 0.001, ** *P* < 0.01.

**Figure 2 cancers-11-00853-f002:**
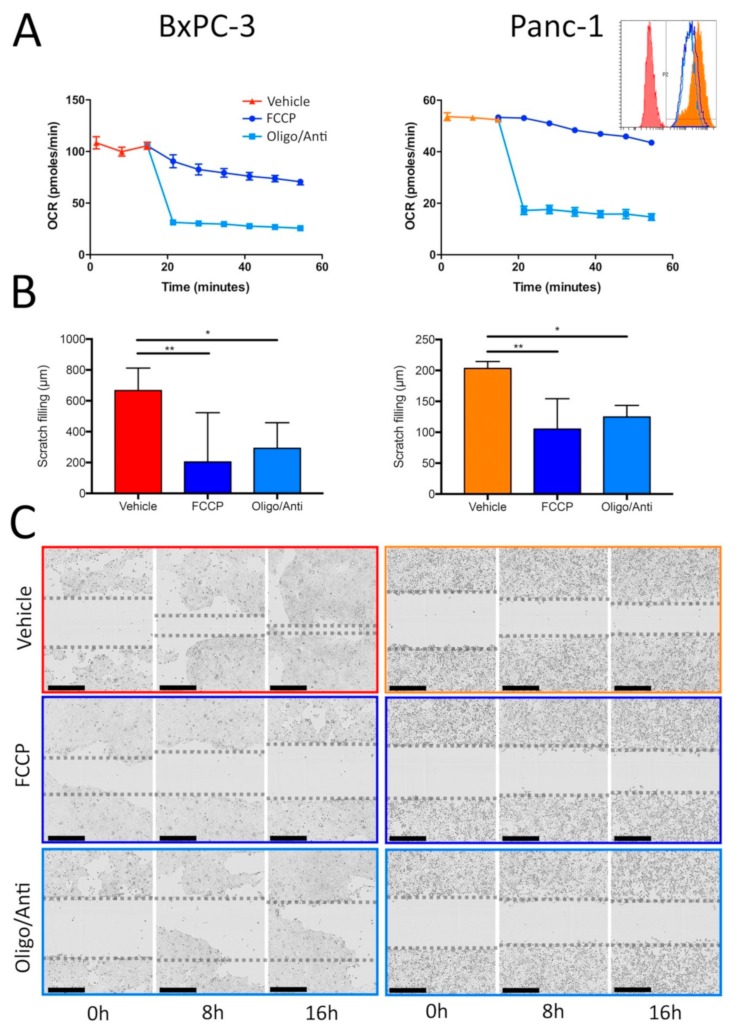
Migration is dependent on oxidative phosphorylation (OXPHOS) in high myoferlin expressing pancreatic ductal adenocarcinoma (PDAC) cell lines. (**A**) Kinetic oxygen consumption rate (OCR) in BxPC-3 and Panc-1 cells before and after treatment with 10 µM FCCP or 10 µM oligomycin (Oligo) + 1 µM antimycin A (Anti). Panc-1 inset represents mitochondrial membrane potential measurement by tetramethylrhodamine, ethyl ester (TMRE) fluorescence in the same conditions. (**B**) End point (16 h) of two-dimension migration kinetic assay (scratch assay) of BxPC-3 and Panc-1 cell lines after treatment with 10 µM FCCP or 10 µM oligomycin (Oligo) + 1 µM antimycin A (Anti). (**C**) Representative images of two-dimension migration kinetic assay with BxPC-3 and Panc-1 after treatment with FCCP 10 µM or 10 µM oligomycin (Oligo) + 1 µM antimycin A (Anti). One representative experiment out of three is illustrated. Each data point represents mean ± SD, n = 3. ** *P* < 0.01, * *P* < 0.05.

**Figure 3 cancers-11-00853-f003:**
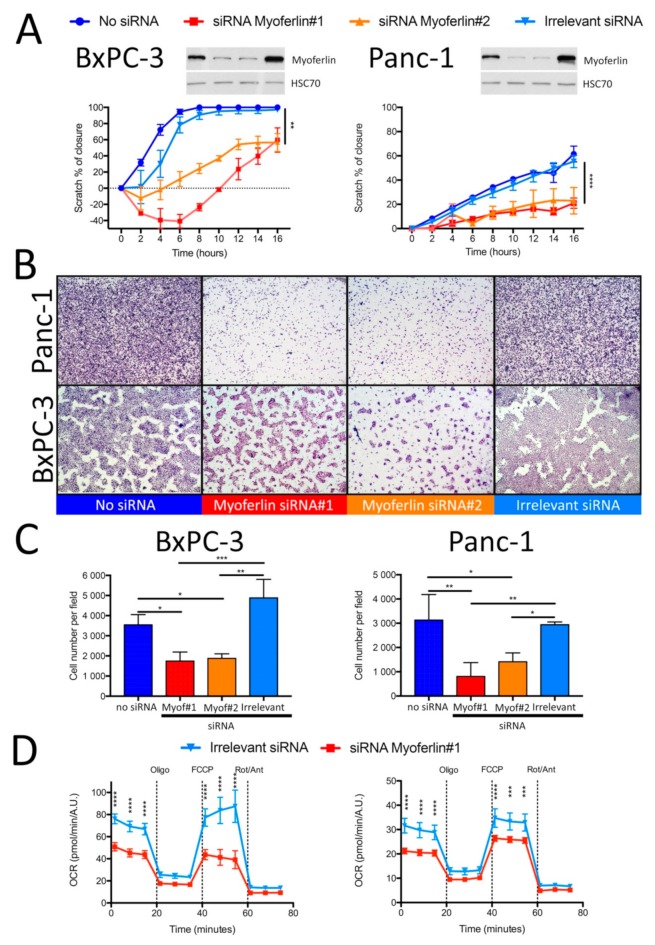
Myoferlin is required for PDAC cell migration and OXPHOS. BxPC-3 and Panc-1 cells were silenced for myoferlin with two different small interfering RNAs (siRNAs). (**A**) Two-dimension migration kinetic assay (scratch assay) of BxPC-3 and Panc-1 cell lines silenced for myoferlin. (**B**) Representative images of 3D migration evaluated in Boyden’s chamber for 24 h. (**C**) Quantification of migrating BxPC-3 and Panc-1 depleted for myoferlin in the lower compartment of Boyden’s chamber. (**D**) Kinetic oxygen consumption rate (OCR) response of irrelevant or myoferlin siRNA-transfected BxPC-3 and Panc-1 cells to oligomycin (oligo, 1 µM), FCCP (1.0 µM), and rotenone and antimycin A mix (Rot/Ant, 0.5 µM each). Upon assay completion, cells were methanol/acetone fixed, and cell number was evaluated using Hoechst incorporation (arbitrary unit, A.U.). One representative experiment out of three is illustrated. Same results were obtained with the second myoferlin siRNA. Each data point represents mean ± SD, n = 3. **** *P* < 0.0001, *** *P* < 0.001, ** *P* < 0.01, * *P* < 0.05.

**Figure 4 cancers-11-00853-f004:**
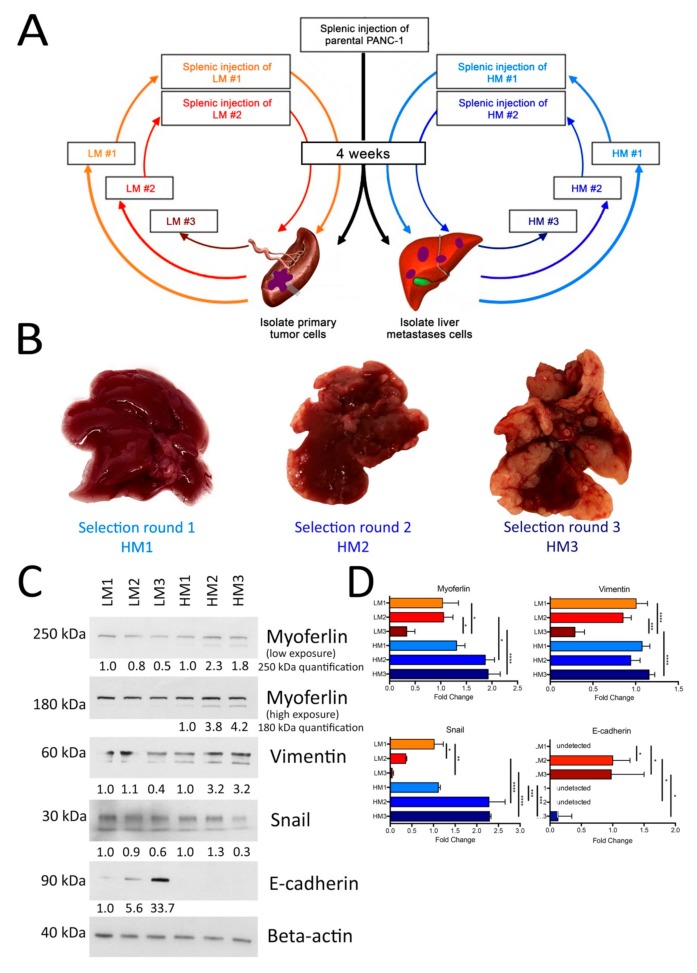
Myoferlin is overexpressed in cells with a high metastatic potential. (**A**) Schematic representation of the in vivo selection of liver-tropic PDAC cells used to generate low metastatic (LM) and highly metastatic (HM) clones. Adapted from [34]. (**B**) Representative liver from mice injected with the liver-tropic Panc-1 cells selected after one, two, or three rounds of injection. (**C**) Western-blot evaluation of myoferlin, vimentin, snail, and E-cadherin abundance in Panc-1 clones selected for their low (LM) or high (HM) metastatic potential. Beta-actin was used as a loading control. (**D**) Gene expression (mRNA) level for myoferlin, vimentin, snail, and E-cadherin in Panc-1 clones selected for their low (LM) or high (HM) metastatic potential. LM1 (LM2 for E-cadherin) gene expression was fixed to 1 and other samples were compared to LM1 (LM2 for E-cadherin). One representative experiment out of three is illustrated. Each data point represents mean ± SD, n = 3. **** *P* < 0.0001, *** *P* < 0.001, ** *P* < 0.01, * *P* < 0.05.

**Figure 5 cancers-11-00853-f005:**
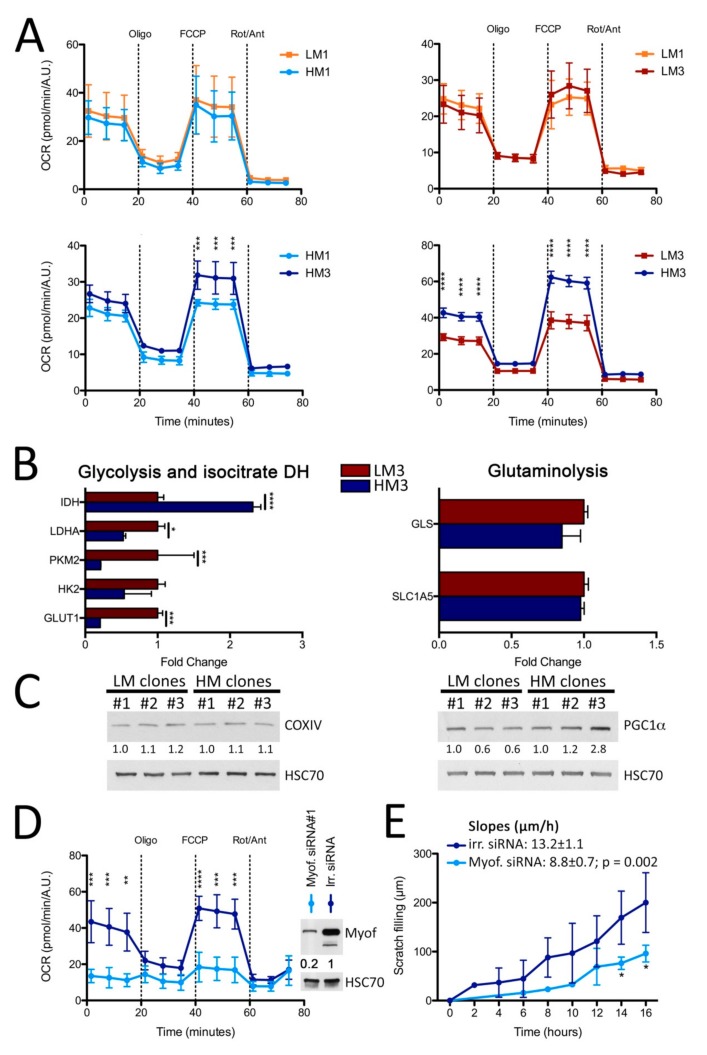
Myoferlin increases OXPHOS activity in HM clones. (**A**) Kinetic oxygen consumption rate (OCR) response of Panc-1 clones (LM1-3 and HM1-3) to oligomycin (oligo, 1 µM), FCCP (1.0 µM), and rotenone and antimycin A mix (Rot/Ant, 0.5 µM each). Upon assay completion, cells were methanol/acetone fixed, and cell number was evaluated using Hoechst incorporation (arbitrary unit, A.U.). (**B**) Metabolism and glutamine-related enzyme gene expression analysis by RT-PCR in LM3 and HM3 Panc-1 clones. (**C**) Cytochrome c oxidase (COX IV) and peroxisome proliferator-activated receptor gamma coactivator 1-alpha (PGC1-α abundance in LM and HM clones. HSC70 was used as a loading control. (**D**) Kinetic oxygen consumption rate (OCR) response of Panc-1 clones (HM3) silenced for myoferlin to oligomycin (oligo, 1 µM), FCCP (1.0 µM), and rotenone and antimycin A mix (Rot/Ant, 0.5 µM each). Upon assay completion, cells were methanol/acetone fixed, and cell number was evaluated using Hoechst incorporation (arbitrary unit, A.U.). (**E**) Two-dimension migration kinetic assay (scratch assay during 16 h) of HM3 Panc-1 clone silenced for myoferlin. One representative experiment out of three is illustrated. Each data point represents mean ± SD, n = 3. **** *P* < 0.0001, *** *P* < 0.001, ** *P* < 0.01, * *P* < 0.05.

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
