# Peer review of "Myoferlin Contributes to the Metastatic Phenotype of Pancreatic Cancer Cells by Enhancing Their Migratory Capacity through the Control of Oxidative Phosphorylation"

_cancers, 2019, doi:10.3390/cancers11060853_

Round 1

Reviewer 1 Report

In this paper, the Authors have generated and used a murine in vivo model to select clones from the human PANC-1 PDAC cell line that exhibit a high propensity to metastasize into the liver.

Specifically they find that myoferlin, overexpressed in PDAC, is involved in the migratory abilities of the selected cells. Moreover, highly PANC-1 metastatic clones expressed significantly higher myoferlin level than low metastatic ones. Finally, they show that myoferlin silencing leads to a migration decrease associated to a reduction of mitochondrial respiration.

I recommend dealing with the following implementations.

Major comments and concerns:

1)   Starting from the reported Fig.1, it seems that some cancer cell lines are unable to invade/migrate. I specifically refer to MiaPaCa-2. While it has been reported that they have migratory ability (Roy et al., 2015 Cancer Research), in this assay they show a different behaviour that needs to be justified.

2)   The data reported in Fig. 3D do not show the effect of siRNAs on OCR. This experiment should be shown.

3)   The effect of myoferlin on EMT was not deeply explored in cell lines experiments. While it seems that it is involved in migration, the molecular characterization of EMT is not reported. The authors should analyse the EMT markers.

4)   It has been reported that E-cadherin expression is under the control of a lncRNA, Hotair, working with Snail to allow gene repression (Battistelli et al., 2018 CDD). Since it has been reported that Hotair is expressed also in PDAC (Kim et al., 2013 Oncogene), the possible involvement of ncRNAs in this regulation should be discussed.

5)   With respect to Fig. 4D, E-cadherin expression should be reported to indicate the inverse correlation between myoferlin and E-cadherin.

Author Response

Reviewer: 1

General Comments:In this paper, the Authors have generated and used a murine in vivo model to select clones from the human PANC-1 PDAC cell line that exhibit a high propensity to metastasize into the liver. Specifically, they find that myoferlin, overexpressed in PDAC, is involved in the migratory abilities of the selected cells. Moreover, highly PANC-1 metastatic clones expressed significantly higher myoferlin level than low metastatic ones. Finally, they show that myoferlin silencing leads to a migration decrease associated to a reduction of mitochondrial respiration.

Specific comments

1. Starting from the reported Fig.1, it seems that some cancer cell lines are unable to invade/migrate. I specifically refer to MiaPaCa-2. While it has been reported that they have migratory ability (Roy et al., 2015 Cancer Research), in this assay they show a different behaviour that needs to be justified.

This reviewer raised a very important point. In our hands, MiaPaCa-2 cell line has a barely detectable myoferlin level and were the slowest to migrate while others reported its significant migratory ability [1]. A major difference in the experimental set-up can explain this apparent discrepancy. Indeed, Roy and coworkers coated the upper well of the Boyden’s chamber to evaluate MiaPaCa-2 migration while our experiments were performed without any coating. There are controversy regarding the migratory ability of PDAC cell lines [2]but recent report indicates a higher EMT potential in Panc-1 than in MiaPaCa-2 [3]. This phenotype specificity can arise from the relative expression of gene regulators such as HOTAIR long non-coding RNA (lncRNA) overexpressed in PANC-1 in comparison to MiaPaCa-2 and associated with enhanced cell invasion [4]and EMT [5]. The Discussion section was amended accordingly.

2. The data reported in Fig. 3D do not show the effect of siRNAs on OCR. This experiment should be shown.

We thank the reviewer for his/her remark indicating that our figure 3D was not adequately designed. Indeed, Figure 3D showed OCR in irrelevant and myoferlin siRNA-transfected BxPC-3 and Panc-1 cell lines. Line color legend was added to the corresponding figure and figure legend was adapted accordingly.

3. The effect of myoferlin on EMT was not deeply explored in cell lines experiments. While it seems that it is involved in migration, the molecular characterization of EMT is not reported. The authors should analyze the EMT markers.

Authors thank the reviewer for his/her suggestion regarding the addition of EMT markers in cell line experiments. We have previously described E-cadherin and vimentin abundance in a panel of PDAC cell lines including BxPC-3, Panc-1, PaTu8988T and MiaPaCa-2 [6], showing E-cadherin expression in BxPC-3 and vimentin expression in Panc-1.According to reviewer suggestion, we have analyzed the abundance of these two EMT markers in myoferlin-silenced BxPC-3 and Panc-1 cell lines. We found that these markers were not affected by myoferlin-silencing, suggesting that the modification of the migratory phenotype observed in cell lines might be the consequence of a metabolic impairment.

4. It has been reported that E-cadherin expression is under the control of a lncRNA, Hotair, working with Snail to allow gene repression (Battistelli et al., 2018 CDD). Since it has been reported that Hotair is expressed also in PDAC (Kim et al., 2013 Oncogene), the possible involvement of ncRNAs in this regulation should be discussed.

Authors thank the reviewer for his/her sounding suggestion improving strongly our manuscript. Discussion section was adapted according to this suggestion. The modification of the migratory phenotype upon myoferlin silencing could arise from the relative expression of gene regulators such as HOTAIR long non-coding RNA (lncRNA) overexpressed in PANC-1 and associated with enhanced cell invasion [4]and EMT [5]. Intriguingly, in a muscle repair context, muscle injuries induce an extensive regulation in coding and noncoding transcripts, among which myoferlin overexpression precedes the one of HOTAIR [7], suggesting a link between these factors or between myoferlin and other non-coding RNA.

5. With respect to Fig. 4D, E-cadherin expression should be reported to indicate the inverse correlation between myoferlin and E-cadherin.

Authors are grateful to the reviewer for his/her suggestion. We performed RT-qPCR for E-cadherin in LM and HM Panc-1 clones. Gene expression data are concordant with the western blot data and correlate negatively with the myoferlin gene expression.

References

1. Roy, I.; McAllister, D. M.; Gorse, E.; Dixon, K.; Piper, C. T.; Zimmerman, N. P.; Getschman, A. E.; Tsai, S.; Engle, D. D.; Evans, D. B.; Volkman, B. F.; Kalyanaraman, B.; Dwinell, M. B. Pancreatic Cancer Cell Migration and Metastasis Is Regulated by Chemokine-Biased Agonism and Bioenergetic Signaling. Cancer Res.2015,75, 3529–3542.

2. Deer, E. L.; González-Hernández, J.; Coursen, J. D.; Shea, J. E.; Ngatia, J.; Scaife, C. L.; Firpo, M. A.; Mulvihill, S. J. Phenotype and genotype of pancreatic cancer cell lines. Pancreas201039, 425–435.

3. Gradiz, R.; Silva, H. C.; Carvalho, L.; Botelho, M. F.; Mota-Pinto, A. MIA PaCa-2 and PANC-1 - pancreas ductal adenocarcinoma cell lines with neuroendocrine differentiation and somatostatin receptors. Sci. Rep.20166, 21648.

4. Kim, K.; Jutooru, I.; Chadalapaka, G.; Johnson, G.; Frank, J.; Burghardt, R.; Kim, S.; Safe, S. HOTAIR is a negative prognostic factor and exhibits pro-oncogenic activity in pancreatic cancer. Oncogene201332, 1616–1625.

5. Battistelli, C.; Sabarese, G.; Santangelo, L.; Montaldo, C.; Gonzalez, F. J.; Tripodi, M.; Cicchini, C. The lncRNA HOTAIR transcription is controlled by HNF4α-induced chromatin topology modulation. Cell Death Differ.2019, 1–12.

6. Rademaker, G.; Hennequière, V.; Brohée, L.; Nokin, M.-J.; Lovinfosse, P.; Durieux, F.; Gofflot, S.; Bellier, J.; Costanza, B.; Herfs, M.; Peiffer, R.; Bettendorff, L.; Deroanne, C.; Thiry, M.; Delvenne, P.; Hustinx, R.; Bellahcène, A.; Castronovo, V.; Peulen, O. J. Myoferlin controls mitochondrial structure and activity in pancreatic ductal adenocarcinoma, and affects tumor aggressiveness.Oncogene201866, 1–15.

7. Aguilar, C. A.; Pop, R.; Shcherbina, A.; Watts, A.; Matheny, R. W.; Cacchiarelli, D.; Han, W. M.; Shin, E.; Nakhai, S. A.; Jang, Y. C.; Carrigan, C. T.; Gifford, C. A.; Kottke, M. A.; Cesana, M.; Lee, J.; Urso, M. L.; Meissner, A. Transcriptional and Chromatin Dynamics of Muscle Regeneration after Severe Trauma. Stem Cell Rep.20167, 983–997.

Reviewer 2 Report

The manuscript “Myoferlin Contributes to the Metastatic Phenotype of Pancreatic Cancer Cells by Enhancing Their Migratory Capacity Through the Control of Oxidative Phosphorylation” by Redemaker et al. The authors found that myoferlin expression is positively associated with the cell migration in pancreatic cancer cells, and that knock-down of myoferlin or treatment with mitochondrial respiratory chain uncoupler FCCP or mitochondrial respiratory chain complex 3 inhibitor antimycin A could suppress the cell migration. The authors collected tissues from primary tumor sites and metastatic sites in mice bearing injected tumor cells and compared the expression levels of Myoferlin, E-Cadherin and Vimentin, and found increased expression of Myoferlin, Vimentin, Snail, and decreased expression of E-Cadherin in selected metastatic cells. The authors claimed that myoferlin increases OXPHOS activity in metastatic tissues, and that knock-down of myoferlin in both primary and metastatic tissue clones would suppress the cells’ response to oligomycin, FCCP or antimycin A. The authors conclude that Myoferlin would promote oxidative phosphorylation to facilitate cell migration. 

Major revision: 

It is unclear to me that for the 2nd (and 3rd round) of injections, are the LM#1 and HM#1 cells mixed and injected together? Based on the fact that the cells are from the same original cell line, and there is no mutagenesis performed, it would be unlikely that the cells would acquire a lot of mutations during the limited cell cycles. It would be helpful for the authors to provide molecular mechanisms associated with those clones, and check what is contributing to the increased expression of myoferlin and OXPHOS activity. Also, if Myoferlin is knocked down in HM3 clone, is the cell migration decreased correspondingly? That could serve as a direct evidence of whether Myoferlin and OXPHOS are required for metastasis, as in LM3 the Myoferlin level is relatively high too (almost equivalent to HM3, Fig5C), while their metastatic potential are expected to be quite different. Also, the Western blotting data and mRNA expression data of Myoferlin are disconcordant in low and high metastatic clones (Fig.4C&D). 

Minor revision

1. Move the full name of FCCP to the first place it appears

2. In Fig3D, the color indication of each line is missing. Is the color labeling same as that in 3A? 

3. Please add Western blotting data of Scale to Fig4C and mRNA data of E-cad to Fig.4D. 

Author Response

Reviewer: 2

General comments: The manuscript “Myoferlin Contributes to the Metastatic Phenotype of Pancreatic Cancer Cells by Enhancing Their Migratory Capacity Through the Control of Oxidative Phosphorylation” by Rademaker et al. The authors found that myoferlin expression is positively associated with the cell migration in pancreatic cancer cells, and that knock-down of myoferlin or treatment with mitochondrial respiratory chain uncoupler FCCP or mitochondrial respiratory chain complex 3 inhibitor antimycin A could suppress the cell migration. The authors collected tissues from primary tumor sites and metastatic sites in mice bearing injected tumor cells and compared the expression levels of Myoferlin, E-Cadherin and Vimentin, and found increased expression of Myoferlin, Vimentin, Snail, and decreased expression of E-Cadherin in selected metastatic cells. The authors claimed that myoferlin increases OXPHOS activity in metastatic tissues, and that knock-down of myoferlin in both primary and metastatic tissue clones would suppress the cells’ response to oligomycin, FCCP or antimycin A. The authors conclude that Myoferlin would promote oxidative phosphorylation to facilitate cell migration. 

Specific comments

1. It is unclear to me that for the 2nd (and 3rd round) of injections, are the LM#1 and HM#1 cells mixed and injected together?

We apologize for the lack of details in the Materials & Methods section. LM and HM cells were injected separately. The Materials & Methods section was amended to explain clearly this point.

2. Based on the fact that the cells are from the same original cell line, and there is no mutagenesis performed, it would be unlikely that the cells would acquire a lot of mutations during the limited cell cycles. It would be helpful for the authors to provide molecular mechanisms associated with those clones, and check what is contributing to the increased expression of myoferlin and OXPHOS activity.

The reviewer raises a pertinent point regarding the establishment of the model used in our study. The in vivo selection of liver-tropic cells by successive rounds of injection in immunocompromised mice is a broadly accepted metastasis model [1-4].This model relies on the observation of tumor or cell line heterogeneity, where highly metastatic cells are present as a subpopulation in primary tumor [5]or cell line, including Panc-1 [6]. The progressive selection of these highly metastatic cells is probably the main driver for the establishment of the described model. This explanation was added to the Discussion section.

3. Also, if Myoferlin is knocked down in HM3 clone, is the cell migration decreased correspondingly? That could serve as a direct evidence of whether Myoferlin and OXPHOS are required for metastasis, as in LM3 the Myoferlin level is relatively high too (almost equivalent to HM3, Fig5C), while their metastatic potential are expected to be quite different.

Authors thank the reviewer for his/her relevant suggestion. The silencing of myoferlin in HM3 clone was performed and 2D migration was assayed. Myoferlin silencing reduced significantly (p=0.002) the migration ability of HM3 clone by >30%. The results were included in the result section and as Fig. 5D. The initial Fig. 5C concerning LM3 was moved to supplemental figure 3B.

We apologize for the quality of our initial Fig. 5C where myoferlin abundance in LM3 clone appeared as high as in HM3 clone. However, western-blots were performed neither on the same membrane nor with the same ECL exposure. As consequence, it is very tricky to compare them. In order to make the myoferlin difference between LM and HM clones more convincing, we add, in Fig. 4C, a low exposure western blot and performed quantifications.

4. Also, the Western blotting data and mRNA expression data of Myoferlin are disconcordant in low and high metastatic clones (Fig.4C&D). 

Authors agree that myoferlin mRNA expression data were apparently discordant with the western blot data. This was due to a voluntary ECL over-exposition in order to show the appearance of the 180 kDa myoferlin isoform. To solve this issue, we have added a lower exposition of the same western blot. The quantification of this western blot is in accordance with the mRNA expression data.

Minor revision

1. Move the full name of FCCP to the first place it appears

The full name of FCCP was moved to the first place it appears.

2. In Fig3D, the color indication of each line is missing. Is the color labeling same as that in 3A?

Reviewer is right, the color labeling of Fig. 3D was the same than in Fig. 3A. A color indication was added in Fig. 3D.

3. Please add Western blotting data of Scale to Fig4C and mRNA data of E-cad to Fig.4D. 

Authors are grateful to the reviewer for his/her suggestion. Western blots in Fig. 4C were quantified. We performed RT-qPCR for E-cadherin in LM and HM Panc-1 clones. Gene expression data are concordant with the western blot data and correlate negatively with the myoferlin gene expression.

References

1. Bruns, C. J.; Harbison, M. T.; Kuniyasu, H.; Eue, I.; Fidler, I. J. In vivo selection and characterization of metastatic variants from human pancreatic adenocarcinoma by using orthotopic implantation in nude mice. Neoplasia19991, 50–62.

2. Price, J. E.; Daniels, L. M.; Campbell, D. E.; Giavazzi, R. Organ distribution of experimental metastases of a human colorectal carcinoma injected in nude mice. Clin. Exp. Metastasis19897, 55–68.

3. Elliott, V. A.; Rychahou, P.; Zaytseva, Y. Y.; Evers, B. M. Activation of c-Met and Upregulation of CD44 Expression Are Associated with the Metastatic Phenotype in the Colorectal Cancer Liver Metastasis Model. PLoS ONE20149, e97432–8.

4. Price, J. E. Spontaneous and experimental metastasis models: nude mice. Methods Mol. Biol.20141070, 223–233.

5. Yokota, J. Tumor progression and metastasis. Carcinogenesis200021, 497–503.

6. Gradiz, R.; Silva, H. C.; Carvalho, L.; Botelho, M. F.; Mota-Pinto, A. MIA PaCa-2 and PANC-1 - pancreas ductal adenocarcinoma cell lines with neuroendocrine differentiation and somatostatin receptors. Sci. Rep.20166, 21648.

Reviewer 3 Report

The authors show that metastatic clones of PDAC expressed higher myoferlin level  than the corresponding low metastatic ones. Human pancreatic cells expressing high myoferlin level have higher migratory potential than cells with a low myoferlin abundance and the migration rate is dependent on OXPHOS activity. Moreover, myoferlin silencing  leads to a migration decrease associated with a reduction of mitochondrial respiration.

The Fig. 1. Panel A seems to be the same reported in fig. 2A of Rademaker et al., Oncogene 2018. The panel should be replaced with another one.

The authors compared the protein level of myoferlin in different human pancreatic cell lines and associated the protein level with the migration rate. However, the cell lines were cultured in different medium, as reported in “methods” section. Can the different culture conditions affect the  myoferlin expression level?

Fig.s 2 and 3. The authors show that the higher migration rate in high myoferlin expressing cell lines is dependent on OXPHOS activity and that the myoferlin silencing induced a reduction of migration and OXPHOS activity. This does not exclude that the same mechanism can occur in cell lines with a lower migration rate and a lower myoferlin level. The same experiments, perhaps at 48 h, should be performed in PaTu8988T and MiaPaCa-2 cell lines. The OXPHOS activity should be also performed in PaTu8988T and MiaPaCa-2 and compared with BxPC-3, 77 Panc-1, even if, again, different culture conditions can affect the OXPHOS activity.

Pag. 9 line 173: Rather than “Myoferlin Increases OXPHOS Activity in HM Clones” Reduction of Myoferlin is associated with a decrease of OXPHOS activity… or the Myoferlin silencing decreases OXPHOS activity …  

Fig. 5, panel B. An evaluation of some OXPHOS component in addition to IDH should be performed.

Fig. 5, panels A and C. The average values of the same clone are different in the panels. For example HM3 has a basal OCR of around 25 in the left panel and 40 in the right panel; the same for LM3 clone comparing panel A and panel C. 

The paper can be strongly improved by evaluating the enzymatic activity of OXPHOS complexes, the aerobic ATP synthesis in basal condition and in myoferlin silencing cell lines.        

In the discussion section the authors could added a speculation on the mechanism by which myoferlin modulates OXPHOS activity.

Author Response

Reviewer: 3

General Comments:The authors show that metastatic clones of PDAC expressed higher myoferlin level than the corresponding low metastatic ones. Human pancreatic cells expressing high myoferlin level have higher migratory potential than cells with a low myoferlin abundance and the migration rate is dependent on OXPHOS activity. Moreover, myoferlin silencing leads to a migration decrease associated with a reduction of mitochondrial respiration.

Specific comments

1. The Fig. 1. Panel A seems to be the same reported in fig. 2A of Rademaker et al., Oncogene 2018. The panel should be replaced with another one

Authors thank the reviewer for his/her careful observation and we apologize for our inattention. The western-blot Fig. 1A was replaced and correlation between migration speed and mean myoferlin abundance was recalculated in accordance.

2. The authors compared the protein level of myoferlin in different human pancreatic cell lines and associated the protein level with the migration rate. However, the cell lines were cultured in different medium, as reported in “methods” section. Can the different culture conditions affect the myoferlin expression level?

The reviewer’s suggestion is very pertinent. Indeed, culture conditions are slightly different between cell lines and are summarized in the table below. Thanks to the supplementation described in the material and methods section, only the glutamine concentration differs from one cell line to the others.

PANC-1

BxPC-3

Patu8988T

MiaPaCa-2

Myoferlin expression

++

+++

+

-

Medium

DMEM

RPMI1640

DMEM

DMEM

Serum

10%

10%

10%

10%

Glucose

4.5 g/L

4.5 g/L

4.5 g/L

4.5 g/L

Glutamine

2 mM

2 mM

2 mM

4 mM

Table 1: Culture conditions for PANC-1, BxPC-3, Patu8988T and MiaPaCa-2 cell lines.

We are currently investigating the impact of culture conditions, mainly glucose and glutamine availability, on myoferlin expression (Fig 1). We noticed that myoferlin expression is only slightly increasing when cells are cultured in low glutamine condition (1mM). As such, we can reasonably assume than the relative myoferlin abundance in the studied cell lines is not the result of culture conditions. For confidential reasons, this information will not be included in the manuscript.

Fig 1: Myoferlin expression in PANC-1 cell line according to culture conditions.

3. Figs 2 and 3. The authors show that the higher migration rate in high myoferlin expressing cell lines is dependent on OXPHOS activity and that the myoferlin silencing induced a reduction of migration and OXPHOS activity. This does not exclude that the same mechanism can occur in cell lines with a lower migration rate and a lower myoferlin level. The same experiments, perhaps at 48 h, should be performed in PaTu8988T and MiaPaCa-2 cell lines. 

We thank the reviewer for his/her very interesting suggestion. However, a 48h migration assay will require the use of an anti-mitotic compound (mitomycin) in order to discriminate the proliferation from the migration. This assay requires first the titration of mitomycin in order to determine its optimal non-toxic concentration. Due to the short deadline (10 days), we feel unable to performed this experiment adequately.

4. The OXPHOS activity should be also performed in PaTu8988T and MiaPaCa-2 and compared with BxPC-3, Panc-1, even if, again, different culture conditions can affect the OXPHOS activity.

Authors thank the reviewer for this relevant comment. Oxygen consumption rate (OCR) of BxPC-3, Panc-1, PaTu8988T and MiaPaCa-2 cell lines were compared on a pairwise fashion in the same culture medium without prior adaptation. Results were included as Fig1E, and showed significant decrease of basal OCR in cell lines according to their low migratory abilities. Interestingly, the basal OCR correlates highly with the myoferlin abundance.

5. Pag. 9 line 173: Rather than “Myoferlin Increases OXPHOS Activity in HM Clones” Reduction of Myoferlin is associated with a decrease of OXPHOS activity… or the Myoferlin silencing decreases OXPHOS activity …

Authors thank the reviewer for the sounding suggestion. Title of section 2.5 was amended according to reviewer’s suggestion.

6. Fig. 5, panel B. An evaluation of some OXPHOS component in addition to IDH should be performed

We want to thank the reviewer for his/her sound comment. We performed a western blot analysis of NADH dehydrogenase subcomplex NDUFB5 (complex 1) and cytochrome C oxidase COXIV (complex 4) in LM1-3 and HM1-3 clones. Unfortunately, NDUFB5 antibody gave us no signal while COXIV showed no significant difference between conditions. COX IV western blot was added as Fig. 5C.

7. Fig. 5, panels A and C. The average values of the same clone are different in the panels. For example HM3 has a basal OCR of around 25 in the left panel and 40 in the right panel; the same for LM3 clone comparing panel A and panel C.

Authors thank the reviewer for his/her relevant remark. Pairwise comparisons of OCR were performed using an XFp seahorse and normalized by Hoechst incorporation as previously described [1]. This normalization procedure is relative and not absolute, explaining why the unit of the OCR axis is pmol/min/A.U. (arbitrary unit). Indeed, the Hoechst incorporation and fluorescence emission are highly depending of several factors including time, temperature, fixation, cell density, … The 4 panels showed in Fig. 5A are biological independent experiments, explaining why the basal OCR levels are different, even in the same clone.

8. The paper can be strongly improved by evaluating the enzymatic activity of OXPHOS complexes, the aerobic ATP synthesis in basal condition and in myoferlin silencing cell lines.

Authors thank the reviewer for his/her sounding suggestion. Unfortunately, our laboratory has not acquired the expertise in performing OXPHOS complex activity yet. As such, this assay was not feasible within the short delay given for this revision. However, we add western blot showing 2 mitochondrial complex proteins in myoferlin-silenced Panc-1 cell line (Supplemental Figure 1D). The myoferlin silencing did not modify the abundance of these proteins. ATP synthesis was previously measured in Panc-1 cell line upon myoferlin silencing and published in Oncogene [1].

9. In the discussion section the authors could added a speculation on the mechanism by which myoferlin modulates OXPHOS activity.

We are grateful to the reviewer for allowing us to expose our hypothesis about the mechanism by which myoferlin controls the mitochondrial function. Of course, we cannot ignore our previous observation demonstrating the disorganization of the mitochondrial network in absence of myoferlin [1]. Obviously, the connection between the bioenergetics and the mitochondrial morphology was previously established [2], but this only moves the problem to determine the link between myoferlin and mitochondrial morphology.As the physiological function of myoferlin is related to membrane biology, it is likely that the mechanism by which myoferlin controls OXPHOS is also related to membrane processes among which we can propose iron uptake or mitochondrial fusion. This aspect was added to the discussion section.

References

1. Rademaker, G.; Hennequière, V.; Brohée, L.; Nokin, M.-J.; Lovinfosse, P.; Durieux, F.; Gofflot, S.; Bellier, J.; Costanza, B.; Herfs, M.; Peiffer, R.; Bettendorff, L.; Deroanne, C.; Thiry, M.; Delvenne, P.; Hustinx, R.; Bellahcène, A.; Castronovo, V.; Peulen, O. J. Myoferlin controls mitochondrial structure and activity in pancreatic ductal adenocarcinoma, and affects tumor aggressiveness.Oncogene201866, 1–15.

2. Alirol, E.; Martinou, J. C. Mitochondria and cancer: is there a morphological connection? Oncogene200625, 4706–4716.

Reviewer 4 Report

In this article the authors showed that myoferlin is overexpressed in highly metastatic pancreatic cells selected through a murine in vivo model. The authors demonstrated that overexpression of  myoferlin is associated with both migratory capability and increased mitochondrial activity of selected cancer cells.

This manuscript is well written and highlights a potential role of myoferlin in therapeutic strategies targeting pancreatic ductal adenocarcinoma (PDAC). However, this article has some weakness that need to be addressed before publication.

1) In the introduction the author should also focus on the pivotal role of tumor microenvironment in cancer metabolic reprogramming (PMID: 29967776, PMID: 27595103).

2) In the line 52 the author should add other references, such as PMID: 31052256, that highlight the role of mitochondrial flexibility in cancer.

3) In the figure 1A the authors should add the densitometric ratio of the means of the three experiments, with relative standard errors and p values. The authors should show the densitometric analysis also in Figure 4 C and in the inset of Figure 5 C.

4) In the line 98 the authors should change "of" with "on".

5) In the subsection 2.2 the authors should demonstrate that the treatments with chain uncoupler  (FCCP) and mitochondrial respiratory chain complex 3 inhibitor (antimycin A) mixed with ATP-103 synthase inhibitor (oligomycin) really impair the mitochondrial functions. To this aim oxygen consumption rate should be measured (PMID: 26534958).

6) The authors should improve the quality of Figure 2B.

7) In Figure 3 the authors should show that the transfection of myoferlin siRNA  is associated with the inhibition of myoferlin synthesis. Therefore, the authors should analyse myoferlin levels by western blotting analysis.

8) In the subsection 2.5, the authors analysed the gene expression of different metabolism-related proteins (Figure 5B). It could be interesting evaluate also PGC1-alpha expression levels, because it represents one of the main positive regulators of OXPHOS (PMID: 12588810). 

Author Response

Reviewer: 4

General Comments:In this article the authors showed that myoferlin is overexpressed in highly metastatic pancreatic cells selected through a murine in vivo model. The authors demonstrated that overexpression of myoferlin is associated with both migratory capability and increased mitochondrial activity of selected cancer cells. This manuscript is well written and highlights a potential role of myoferlin in therapeutic strategies targeting pancreatic ductal adenocarcinoma (PDAC). However, this article has some weakness that need to be addressed before publication.

Specific comments

1. In the introduction the author should also focus on the pivotal role of tumor microenvironment in cancer metabolic reprogramming (PMID: 29967776, PMID: 27595103).

Authors thank the reviewer for his/her relevant suggestion. We have added some information in the introduction section to highlight the importance of the “reverse Warburg effect” and we have cited Arcucci’s reviews as suggested, as well as Sousa’s article describing the alanine feeding of PDAC cells by stellate cells.

2. In the line 52 the author should add other references, such as PMID: 31052256, that highlight the role of mitochondrial flexibility in cancer.

As requested by the reviewer, the review authored by Avagliano and coworkers was cited in the context of the mitochondrial flexibility.

3. In the figure 1A the authors should add the densitometric ratio of the means of the three experiments, with relative standard errors and p values. 

Authors thank the reviewer for his/her sounding remark improving our results. Densitometric analysis was performed in 3 independent biological replicates. Mean±sem, as well as p-value from one-sample T test, were added in Fig. 1A. Results confirmed the myoferlin expression difference between Panc-1, Patu8988T and MiaPaCa-2. Correlation between migration speed and mean myoferlin expression was recalculated.

4. The authors should show the densitometric analysis in Figure 4 C and in the inset of Figure 5 C.

Authors thank for the reviewer for his/her sounding suggestion improving the information delivered by our results. Densitometric analysis were included in Fig. 4C. and in Fig 5C insets.

5. In the line 98 the authors should change "of" with "on".

We thank the reviewer for his/her careful proofreading. The sentence was corrected as required.

6. In the subsection 2.2 the authors should demonstrate that the treatments with chain uncoupler (FCCP) and mitochondrial respiratory chain complex 3 inhibitor (antimycin A) mixed with ATP synthase inhibitor (oligomycin) really impair the mitochondrial functions. To this aim oxygen consumption rate should be measured (PMID: 26534958).

We want to thank the reviewer for having suggest us this interesting control. Although, FCCP or Antimycin + Oligomycin treatments were described previously by Allen and coworker [1]in EMBO report for inducing mitochondrial damage leading to mitophagy, we have performed an oxygen consumption rate (OCR) measurement in presence of these compounds. Panc-1 and BxPC-3 showed a sharp decrease of their OCR in presence of Antimycin/Oligomycin, indicating a strong impairment of the respiratory complexes. FCCP alone produced a slow decrease of the OCR, as described by Figarola and coworker with higher concentration of FCCP (25 µM) [2], indicating that the uncoupling activity of the compound is not able anymore to drive mitochondria to maximal OCR. Since this decrease was slower in Panc-1 cells than in BxPC-3, we decide to analyze the mitochondrial potential of Panc-1 after Antimycin + Oligomycin or FCCPtreatments. Both conditions reduced the TMRE fluorescence, indicating a loss of the mitochondrial potential and confirming the impairment of the mitochondrial function. These results were added to the manuscript as Fig. 2A.

7. The authors should improve the quality of Figure 2B.

Authors apologize for the poor quality of Figure 2B. The contrast of each image was adjusted and normalized in order to increase the overall quality without tampering the results.

8. In Figure 3 the authors should show that the transfection of myoferlin siRNA is associated with the inhibition of myoferlin synthesis. Therefore, the authors should analyze myoferlin levels by western blotting analysis.

Author thank the reviewer for this suggestion. In Fig 3A, we have added the western-blot showing the siRNA transfection efficiency of the concordant migration assay.

9. In the subsection 2.5, the authors analyzed the gene expression of different metabolism-related proteins (Figure 5B). It could be interesting evaluate also PGC1-alpha expression levels, because it represents one of the main positive regulators of OXPHOS (PMID: 12588810). 

Authors thank the reviewer for his/her sounding suggestion improving strongly our manuscript. PGC1-a is broadly described as a positive regulator of mitochondrial function and its expression level was checked by western blot in LM and HM clones. We observed a slight decrease of PGC1a in LM clones according to the injection round, and an increase of its abundance in HM clones following their migratory potential. This point confirms the metabolic switch occurring in migratory PDAC cells.

References

1. Allen, G. F. G.; Toth, R.; James, J.; Ganley, I. G. Loss of iron triggers PINK1/Parkin-independent mitophagy. EMBO Rep.201314, 1127–1135.

2. Figarola, J. L.; Singhal, J.; Tompkins, J. D.; Rogers, G. W.; Warden, C.; Horne, D.; Riggs, A. D.; Awasthi, S.; Singhal, S. S. SR4 uncouples mitochondrial oxidative phosphorylation, modulates AMP-dependent Kinase (AMPK)-mammalian target of rapamycin (mTOR) signaling, and inhibits proliferation of HepG2 hepatocarcinoma cells. J. Biol. Chem.2015,290, 30321–30341.

Round 2

Reviewer 1 Report

The manuscript is suitable for publication in the present form

Author Response

Authors want to thank the reviewer for his/her comments. We are delighted that our work was judged as publishable.

Reviewer 2 Report

The modifications are satisfactory to the concerns regarding the previous version of manuscript. 

Minor revision: 

1. Please add the western blotting data of SNAIL expression to Fig.4C. 

Author Response

Minor revision - comment 1 : Please add the western blotting data of SNAIL expression to Fig.4C.

Authors are grateful to the reviewer for his/her pertinent suggestion. Snail western blot was added to Fig. 4C and quantified. Results were described in the result section.

Reviewer 3 Report

The authors have addressed my concerns

Author Response

Authors want to thank reviewer for his/her comments regarding our revision. We are delighted that our revision was judged as publishable.

Reviewer 4 Report

The paper has been substantially improved by the authors, but there are still some major points left:

The  densitometric analysis of western blotting should always be associated with a graph showing  the mean of three independent experiments, with relative standard deviation and p value. Therefore, the authors  should add to Figures  4 C  and 5 C  the  densitometric analysis of western blotting with a graph showing  the mean of three independent experiments, with relative standard deviation and p value. 

Author Response

Major point - The  densitometric analysis of western blotting should always be associated with a graph showing  the mean of three independent experiments, with relative standard deviation and value. Therefore, the authors  should add to Figures  4 C  and 5 C  the  densitometric analysis of western blotting with a graph showing  the mean of three independent experiments, with relative standard deviation and value.

Authors thank the reviewer for his/her sound remarks. Western-blot data from three independent experiments were compiled. For reason of space, the densitometric analysis were shown in supplemental figure 2 and 3 as mean with standard deviation. Due to homoscedasticity suspicion, statistical analysis were performed using Kruskall-Wallis one-way ANOVA, followed by a Dunn's post-test.

Round 3

Reviewer 4 Report

The paper has been substantially improved by the authors, therefore it can be accepted in present form.